# EQUILIBRIUM FINDING VIA EXPLOITABILITY DESCENT WITH LEARNED BEST-RESPONSE FUNCTIONS

## ABSTRACT

There has been great progress on equilibrium finding research over the last 20 years. Most of that work has focused on games with finite, discrete action spaces. However, many games involving space, time, money, *etc.* have continuous action spaces. We study the problem of computing approximate Nash equilibria of games with continuous strategy sets. The main measure of closeness to Nash equilibrium is *exploitability*, which measures how much players can benefit from unilaterally changing their strategy. We propose a new method that minimizes an approximation of exploitability with respect to the strategy profile. This approximation is computed using *learned best-response functions*, which take the current strategy profile as input and return learned best responses. The strategy profile and best-response functions are trained simultaneously, with the former trying to minimize exploitability while the latter try to maximize it. We evaluate our method on various continuous games, showing that it outperforms prior methods.

## 1 INTRODUCTION

Most work concerning equilibrium computation has focused on games with finite, discrete action spaces. However, many games involving space, time, money, *etc.* have continuous action spaces. Examples include continuous resource allocation games (Ganzfried, 2021), security games in continuous spaces (Kamra et al., 2017; 2018; 2019), network games (Ghosh & Kundu, 2019), military simulations and wargames (Marchesi et al., 2020), and video games (Berner et al., 2019; Vinyals et al., 2019). Moreover, even if the action space is discrete, it can be fine-grained enough to treat as continuous for the purpose of computational efficiency Borel (1938); Chen & Ankenman (2006); Ganzfried & Sandholm (2010b).

The first question that arises in the multiagent setting is what game-theoretic solution concept and performance metric should be used. For the former, we use the standard solution concept of a Nash equilibrium: that is, a strategy profile for which each strategy is a best response to the remaining players' strategies. The main measure of closeness to Nash equilibrium is *exploitability*, which measures how much players can benefit from unilaterally changing their strategy. Typically, we seek Nash equilibria, that is, strategy profiles for which the exploitability is zero. As some previous works in the literature have done, we can try to search for Nash equilibria by performing gradient descent on exploitability, since it is non-negative and its zero set is precisely the set of Nash equilibria. However, evaluating exploitability requires computing best responses to the current strategy profile, which is itself a nontrivial problem in complex games.

We propose a new method that minimizes an approximation of the exploitability with respect to the strategy profile. This approximation is computed using *learned best-response functions*, which take the current strategy profile as input and return learned best responses. The strategy profile and best-response functions are trained simultaneously, with the former trying to minimize exploitability while the latter try to maximize it.

We start by introducing some background needed to formulate the problem, including the definition of strategic-form games, strategies, equilibria, and exploitability. Next, we describe some prior methods in the literature and related research. We then describe various games that we use as benchmarks and discuss our experimental results. Finally, we present our conclusion and suggest directions for future research.

## 2 BACKGROUND

**Games and equilibria**  A *strategic-form game* is a tuple $(I, X, u)$ where $I$ is a set of players, $X_i$ is a strategy set for player $i$, and $u : \prod_i X_i \to \mathbb{R}^I$ is a utility function. A strategy profile $x \in \prod_i X_i$ maps each player to a strategy for that player. A game is zero-sum if and only if $\sum_i u(x)_i = 0$ for all strategy profiles $x$. $x_{-i}$ denotes $x$ excluding player $i$'s strategy. Player $i$'s best-response utility $b(x)_i = \sup_{x'_i} u(x'_i, x_{-i})_i$ is the highest utility they can attain given the other players' strategies. Their utility gap $\delta(x)_i = b(x)_i - u(x)_i$ is the highest utility they can gain from unilaterally changing their strategy. $x$ is an $\varepsilon$-equilibrium iff $\sup_i \delta(x)_i \leq \varepsilon$. A 0-equilibrium is called a Nash equilibrium. In a Nash equilibrium, each player's strategy is a best response to the other players' strategies, that is, $u(x)_i \geq u(x'_i, x_{-i})_i$ for all $i \in I$ and $x'_i \in X_i$.

**Infinite games**  For some games, the $X_i$ might be infinite. The following theorems apply to such games: If for all $i$, $X_i$ is nonempty and compact, and $u(x)_i$ is continuous in $x$, a mixed strategy Nash equilibrium exists (Glicksberg, 1952). If for all $i$, $X_i$ is nonempty, compact, and convex, and $u(x)_i$ is continuous in $x$ and quasi-concave in $x_i$, a pure strategy Nash equilibrium exists (Fudenberg & Tirole, 1991, p. 34). Other results include the existence of a mixed strategy Nash equilibrium for games with discontinuous utilities under some mild semicontinuity conditions on the utility functions (Dasgupta & Maskin, 1986), and the uniqueness of a pure Nash equilibrium for continuous games under diagonal strict concavity assumptions (Rosen, 1965).

**Nikaido-Isoda function**  Nikaido & Isoda (1955) introduced the *Nikaido-Isoda (NI)* function $\phi(x, y) = \sum_i (u(y_i, x_{-i})_i - u(x)_i)$. It is also sometimes called the Ky Fan function (Flåm & Antipin, 1996; Flåm & Ruszczyński, 2008; Hou et al., 2018). Several papers have proposed algorithms that use this function to find Nash equilibria, including Berridge & Krawczyk (1970); Uryasev & Rubinstein (1994); Krawczyk & Uryasev (2000); Krawczyk (2005); Flåm & Ruszczyński (2008); Gürkan & Pang (2009); von Heusinger & Kanzow (2009a;b); Qu & Zhao (2013); Hou et al. (2018); Raghunathan et al. (2019); Tsaknakis & Hong (2021).

**Exploitability**  Let $\psi(x) = \sup_y \phi(x, y)$. Then $\psi(x) \geq 0$. Furthermore, $\psi(x) = 0$ if and only if $x$ is a Nash equilibrium. $\psi$ is commonly known as the *exploitability* or *Nash convergence metric (NashConv)* in the literature (Lanctot et al., 2017; Lockhart et al., 2019; Walton & Lisy, 2021; Timbers et al., 2022). It adds up the utility gaps $\delta(x)_i = \sup_{y_i} u(y_i, x_{-i})_i - u(x)_i$ of each player, and thus serves as a measure of closeness to Nash equilibrium. In a two-player zero-sum game, the exploitability reduces to the so-called "duality gap" (Grnarova et al., 2021) $\psi(x) = \sup_{x'_1} u(x'_1, x_2)_1 - \inf_{x'_2} u(x_1, x'_2)_1$.

**Algorithms**  There are several algorithms in the literature for continuous games. Mescheder et al. (2017) introduced *consensus optimization (CO)* to improve the convergence properties of GANs. Unfortunately, CO can sometimes converge to undesirable points even in simple potential games. Balduzzi et al. (2018) introduced *symplectic gradient adjustment (SGA)* to address some of the shortcomings of CO.

Foerster et al. (2018) introduced *learning with opponent-learning awareness (LOLA)*, a method in which each agent shapes the anticipated learning of the other agents in the environment. The LOLA learning rule includes a term that accounts for the impact of one agent's policy on the anticipated parameter update of the other agents. They show that LOLA leads to cooperation with high social welfare, while independent policy gradients, a standard multiagent RL approach, does not. The policy gradient finding is consistent with prior work, such as Sandholm & Crites (1996). Letcher et al. (2018) introduced *stable opponent shaping (SOS)*, which interpolates between LOLA and a different but similar learning method called *LookAhead* (Zhang & Lesser, 2010).

Schaefer & Anandkumar (2019) introduced *competitive gradient descent (CGD)*, a natural generalization of gradient descent to the two-player setting where the update is given by the Nash equilibrium of a regularized bilinear local approximation of the underlying game. It avoids oscillatory and divergent behaviors seen in alternating gradient descent. The convergence and stability properties of their method are robust to strong interactions between the players, without adapting the stepsize, which is not the case with previous methods. Ma et al. (2021) generalized CGD to more than two players by using a local approximation given by a multilinear polymatrix game that can

be solved using linear algebraic methods. They call this method *polymatrix competitive gradient descent (PCGD)*.

To describe these and other methods in the literature formally, we use the following notation. Let $\xi = \operatorname{diag} \nabla u$ be the *simultaneous gradient*; each component $\xi_i = \nabla_i u_i$ is the gradient of a player's utility with respect to their strategy. It is a vector field on the strategy profile space. It need not be conservative (the gradient of some potential), which is the primary source of difficulties in applying standard gradient-based optimization methods, since trajectories can cycle around fixed points rather than converge to them. Let $H = \nabla \xi$. Then *naive learning (NL)* (also known as simultaneous gradient ascent), *extragradient (EG)* (Korpelevich, 1976), *optimism (OP)* (Popov, 1980; Daskalakis et al., 2018), *consensus optimization (CO)* (Mescheder et al., 2017), *sympletic gradient adjustment (SGA)* (Balduzzi et al., 2018), *lookahead (LA)* (Zhang & Lesser, 2010), *symmetric lookahead (SLA)* (Letcher, 2018), *learning with opponent-learning awareness (LOLA)* (Foerster et al., 2018), *gradient-based Nikaido-Isoda (GNI)* (Raghunathan et al., 2019), and *polymatrix competitive gradient descent (PCGD)* (Ma et al., 2021) are, respectively:

$$\dot{x}_{\text{NL}} = \xi \tag{1}$$

$$\dot{x}_{\text{EG}} = \xi|_{x+\gamma\xi} \tag{2}$$

$$\dot{x}_{\text{OP}} = \xi + \gamma\dot{\xi} \tag{3}$$

$$\dot{x}_{\text{CO}} = (I - \gamma H^{\mathsf{T}})\xi = \xi - \gamma\nabla\tfrac{1}{2}\|\xi\|^2 \tag{4}$$

$$\dot{x}_{\text{SGA}} = (I - \gamma H_a^{\mathsf{T}})\xi \tag{5}$$

$$\dot{x}_{\text{LA}} = (I + \gamma H_o)\xi \tag{6}$$

$$\dot{x}_{\text{SLA}} = (I + \gamma H)\xi \tag{7}$$

$$\dot{x}_{\text{LOLA}} = \dot{x}_{\text{LA}} - \gamma\operatorname{diag} H_o^{\mathsf{T}}\nabla u \tag{8}$$

$$\dot{x}_{\text{GNI}} = -\nabla\phi(x, x + \gamma\xi) \tag{9}$$

$$\dot{x}_{\text{PCGD}} = (I - \gamma H_o)^{-1}\xi \tag{10}$$

where $\gamma > 0$ is a hyperparameter, $\xi|_y$ is $\xi$ evaluated at $y$ rather than $x$, $H^{\mathsf{T}}$ is the transpose of $H$, $H_a = \frac{1}{2}(H - H^{\mathsf{T}})$ is the antisymmetric part of $H$, and $H_o$ is the off-diagonal part of $H$ (replacing its diagonal with zeroes). GNI replaces the supremum in $\psi$ with the local approximation $y = x + \gamma\xi$. SLA is a linearized version of EG (Enrich, 2019, Lemma 1.35). LA is the series expansion of PCGD to first order in $\gamma$ (Willi et al., 2022, Proposition 4.4), since $(I - \gamma M)^{-1} = I + \gamma M + \gamma^2 M^2 + \dots$ for sufficiently small $\gamma$. Exact PCGD requires solving a linear system of equations, which makes each iteration costly (Ma et al., 2021, p. 10). The actual optimization is done by discretizing each ODE in time. For example, with NL, we have the iteration scheme $x_{i+1} = x_i + \eta\xi_i$ for some small learning rate $\eta > 0$. With OP, we have $x_{i+1} = x_i + \eta\xi_i + \gamma(\xi_i - \xi_{i-1})$.

These algorithms have been analyzed in various works, including Balduzzi et al. (2018); Letcher et al. (2018; 2019); Mertikopoulos & Zhou (2019); Grnarova et al. (2019); Mazumdar et al. (2019); Hsieh et al. (2021); Willi et al. (2022). Additional related research can be found in the appendix.

## 3 OUR METHOD

Before introducing our method, we motivate it by considering the perspective of a single player in a two-player game, with the aim of recommending a "good strategy" for that player.

**Single-player perspective** Let $X, Y$ be sets and $u : X \times Y \to \mathbb{R}$ be a differentiable function. Suppose we want to solve the following optimization problem: $\max_x \min_y u(x, y)$. This corresponds to finding a minimally-exploitable strategy for Player 1 in a zero-sum game with payoff function $u$. To solve this problem, we could try to apply simultaneous gradient ascent:

$$\dot{x} = +\nabla_x u(x, y) \qquad \dot{y} = -\nabla_y u(x, y) \tag{11}$$

Unfortunately, this approach can easily fail even in simple games. For example, consider the simple bilinear game that has $X, Y \subset \mathbb{R}$ and $u(x, y) = xy$. $x$ should converge to 0, but instead oscillates around it indefinitely. In fact, $x$ and $y$ jointly trace a circle around the origin. The essence of this cycling problem is that Player 2 has to "relearn" a good response to Player 1 every time the latter's

strategy switches sign. This is a general problem for games: small changes in other players' strategies may cause big changes in a player's best response. Thus players have to essentially "relearn" how to respond to the other players' strategies every time such discontinuous changes occur.

**Best-response functions** To address this, we can reformulate the problem as maximizing $u(x, b(x))$ with respect to $x$, where $b : X \to Y$ is a *function* that satisfies $b(x) \in \arg\min_y u(x, y)$. Since $b$ is a fully-fledged *function*, it can map different strategies for Player 1 to different strategies for Player 2. Thus it can *immediately* adapt to Player 1's strategy and avoid the cycling problem. To find $x$ and such a $b$ simultaneously, we can perform simultaneous gradient ascent:

$$\dot{x} = +\nabla_x u(x, b(x)) \qquad \dot{b} = -\nabla_b u(x, b(x)) \tag{12}$$

This time, we use the *functional derivative* $\nabla_b$. Again, since $b$ is a fully-fledged function, Player 1's changing behavior poses no fundamental hindrance to it learning good responses, and "saving" them for later use if Player 1's behavior changes. It could even learn the true best-response function and stop there, leaving Player 1 to face a simple standard optimization problem.

If $X$ is infinite and $Y$ is nontrivial, $X \to Y$ is infinite-dimensional. To represent and optimize $b$ in practice, we need a finite-dimensional parameterization of (a subset of) this function space. We use neural networks for this, letting $b$ be the function defined by a neural network with a given set of parameters, and optimize these parameters.

$X$ itself may be a function space. For example, Player 1's strategy may be a policy that takes observations as input and outputs actions accordingly. In that case, $b$ acts as a *higher-order function* that takes a function (Player 1's strategy) as input and returns a new function (a strategy for Player 2). In practice, it can simply take the *parameters* of Player 1's policy as input.

**Approach for n-player games** We can try to search for Nash equilibria by performing gradient descent on the exploitability function $\psi$, since it is non-negative and its zero set is precisely the set of Nash equilibria. However, computing the supremum for $\psi$ is expensive and/or intractable in the general case, since it requires computing best responses to the current strategy profile, itself a nontrivial problem. Inspired by our approach for the 2-player setting, we reformulate $\psi$ as $\psi(x) = \phi(x, b(x))$ where $b$ is a *best-response function* such that $b(x)_i \in \arg\max_{x_i'} u(x_i', x_{-i})_i$. In practice, we model best-response functions with neural networks, which have a finite-dimensional parameterization, and train them simultaneously with $x$. Thus our scheme is:

$$\dot{x} = -\nabla_x \phi(x, b(x)) \qquad \dot{b} = +\nabla_b \phi(x, b(x)) \tag{13}$$

That is, the best responses try to *increase* the exploitability while the strategies try to *decrease* it. More precisely, if $b$ is parameterized by a finite-dimensional parameter vector $\theta$, then our scheme is:

$$\dot{x} = -\nabla_x \phi(x, b_\theta(x)) \qquad \dot{\theta} = +\nabla_\theta \phi(x, b_\theta(x)) \tag{14}$$

We call our method *exploitability descent with learned best-response functions*.

## 4 EXPERIMENTS

In our experiments, we use *affine* best-response functions of the form $b_\theta(x) = Ax + b$ where $(A, b) = \theta$. We initialize $\theta$ so that $b_\theta$ is the identity function. Unless otherwise specified, we use a learning rate of $10^{-2}$ and $\gamma = 10^{-1}$. For the algorithms that use $\gamma$, lower values of $\gamma$ slowed their convergence rate, bringing it closer to that of NL. Higher values of $\gamma$ accelerated convergence on some but not all games. Hyperparameter comparisons can be found in the appendix.

Some of these benchmarks are based on normal-form or extensive-form games with finite action sets, and thus finite-dimensional continuous mixed strategies. While there are algorithms for such games that might have better performance (such as counterfactual regret minimization (Zinkevich et al., 2007) and its variants), these do not readily generalize to general continuous-action games. Thus we are interested in comparing only to those algorithms which, like ours, *do* generalize to continuous-action games, namely those described in the previous section.

We parameterize mixed strategies on a finite action set (*e.g.*, at a particular information set inside an extensive-form game) as follows. Suppose there are $n + 1$ actions. Then we use a parameter $x \in \mathbb{R}^n$ and define the action probabilities as softmax(append($x, 0$)). When $x$ is 1-dimensional, this is equivalent to applying the logistic sigmoid function.

**Sign game** This game has $n$ players, $X_i = [-100, 100]$, and

$$u(x)_i = \begin{cases} x_i x_{i+1} & i < n \\ -x_n x_1 & i = n \end{cases} \tag{15}$$

This is an $n$-player generalization of the simple bilinear zero-sum game $u(x, y)_1 = -u(x, y)_2 = xy$. The latter is an important example because it captures the cycling behavior of simultaneous learning dynamics when training GANs (Mescheder et al., 2018; Zhang & Yu, 2020). This game has an equilibrium at the origin.

The performance of each algorithm is illustrated in Figure 2a (for 2 players) and Figure 2b (for 3 players). The trajectories of each player's parameter for the 2-player case are shown in Figure 1a. NL slowly spirals away from the equilibrium. All algorithms from OP to LA (inclusive) overlap exactly and slowly spiral into the equilibrium. LOLA also spirals into the equilibrium, but faster. Our algorithm shoots straight for the equilibrium, overshoots it a bit, and bounces back along the same line, getting closer and closer to the equilibrium with each oscillation. It converges significantly faster than the other algorithms.

**Antisymmetric game** This game has 4 players, $X_i = [-100, 100]$, and $u(x)_i = x_i \sum_j x_j \operatorname{sign}(j - i)$. It appears in Balduzzi et al. (2018, Appendix D). The performance of each algorithm on this game is illustrated in Figure 2c. On this game, LOLA and our algorithm converge significantly faster than the rest, with the former converging slightly faster.

**Pennies game** This is the strategic-form version of a normal-form game with $n$ players, action sets $A_i = \{H, T\}$, and payoff function

$$u(a)_i = \begin{cases} [a_i = a_{i+1}] & i < n \\ [a_n \neq a_1] & i = n \end{cases} \tag{16}$$

where $[\cdot]$ denotes the Iverson bracket, which is 1 if its argument is true and 0 otherwise. This is a generalization of the classic *matching pennies game* to $n$ players (Jordan, 1993; Leslie & Collins, 2003). Each player but the last tries to match the next player, and the last player tries to *un*match the first. Like the 2-player version, it has a unique mixed strategy equilibrium where each player mixes uniformly between heads and tails. The payoff matrix for $n = 2$ is shown in Table 1a.

Interestingly, Jordan (1993) shows that the 3-player game's Nash equilibrium is locally unstable in a strong sense: for any $\varepsilon > 0$, and for almost all initial empirical distributions that are within Euclidean distance $\varepsilon$ of the equilibrium, discrete-time fictitious play does not converge to the equilibrium. Instead, it enters a limit cycle asymptotically as $t \to \infty$.

We can construct a strategic-form game whose strategy sets are the mixed strategies of this normal-form game, and whose payoffs are the expected payoffs of the normal-form game under such mixed strategies. We use a softmax parameterization of the strategies, as described in the beginning of this section. The game then has a unique equilibrium at the origin.

The performance of each algorithm is illustrated in Figure 2d (for 2 players) and Figure 2e (for 3 players). The trajectories of each player's parameter for the 2-player case are shown in Figure 1b. NL slowly spirals away from the equilibrium. All algorithms from SLA to CO (inclusive) overlap almost exactly and slowly spiral into the equilibrium. LOLA also spirals into the equilibrium, but slightly faster. Our algorithm shoots straight for the equilibrium, overshoots it, and bounces back, getting closer and closer to the equilibrium with each oscillation. It converges significantly faster.

**Rock paper scissors game** This is the strategic-form version of the classic rock paper scissors game, a 2-player normal-form game whose payoff matrix is shown in Table 1b. The equilibrium lies at the origin. The performance of each algorithm is illustrated in Figure 2f. The trajectories of each player's action probabilities (a point on the 2-simplex) are illustrated in Figure 4a in the appendix. Our algorithm converges faster than the rest, with LOLA attaining the next-best performance.

**Shapley game** This is the strategic-form version of the Shapley game, a normal-form game whose payoff matrix is shown in Table 1c. It was introduced by Shapley (1964, p. 26) and is a well-known

example of a game for which fictitious play does not converge. Instead, fictitious play cycles through the cells with 1's in them, with ever-increasing lengths of play in each of these cells.

The performance of each algorithm is illustrated in Figure 2g. The trajectories of each player's action probabilities are illustrated in Figure 4b in the appendix. The difference in performance between our algorithm and the others is especially stark in this game. All other algorithms seem to diverge from the equilibrium quite rapidly, whereas ours converges rapidly toward it.

**Glicksberg–Gross game**   This is the strategic form version of a 2-player *continuous*-action normal-form game with action sets $A_i = [0, 1]$ and payoff function

$$u(x, y)_1 = -u(x, y)_2 = \frac{(1 + x)(1 + y)(1 - xy)}{(1 + xy)^2} \tag{17}$$

This is a 2-player zero-sum game on the unit square. It was studied by Glicksberg & Gross (1953), who showed that it has a *unique* mixed strategy Nash equilibrium with the probability density function $f(x) = \frac{2}{\pi\sqrt{x}(1+x)}$ for both players. As the authors comment, this shows that rational games do not share with polynomial games the property of always having step function solutions.

For this game, we model players' strategies as beta distributions that are parameterized as follows: Each player's strategy is a pair of real numbers. These are passed through a softplus function to make them positive, and the results are used as the concentration parameters. Due to the variance of our stochastic estimator for the game's utility function, we use a learning rate of $10^{-4}$. Figure 2h illustrates the performance of each algorithm. Ours attains a lower exploitability than the other algorithms and appears to have lower variance. Figure 3a in the appendix illustrates the probability density functions of each player after training.

**Circle game**   This is the strategic-form version of an $n$-player continuous-action normal-form game with action sets $A_i = [-\pi, \pi)$ and payoff function

$$u(a) = \begin{cases} -d(a_i, a_{i+1}) & i < n \\ d(a_n, a_1) & i = n \end{cases} \tag{18}$$

where $d(x, y)$ is the Euclidean distance between the points at angles $x$ and $y$ on the unit circle. Each player chases the next, but the last avoids the first. This is a generalization of the 2-player zero-sum distance game on the unit circle. Like the 2-player version, it has a mixed strategy equilibrium where each player mixes uniformly on the unit circle.

We let player strategies be von Mises distributions (of which the circular uniform distribution is a special case) that are parameterized as follows: Let $x_i \in \mathbb{R}^2$ be Player $i$'s parameter. Then $\mu = \arctan2(x_i) \in [-\pi, \pi)$ is the angular position parameter and $\kappa = \|x_i\| \in [0, \infty)$ is the angular scale parameter, where arctan2 yields the angle of its input (in radians). Due to the variance of our stochastic estimator for the game's utility function, we use a learning rate of $10^{-4}$. Figure 2i and Figure 2j illustrate the performance of each algorithm for 2 players and 3 players, respectively. Our algorithm attains a lower exploitability and variance than the rest. Figure 3b in the appendix illustrates the probability density functions of each player after training.

**Kuhn poker**   This is a famous simplified version of poker introduced by Kuhn (1950) as a model of zero-sum two-player imperfect-information games that can be analyzed theoretically. A 3-player variant was introduced by Szafron et al. (2013), who stated that it was one of the largest three-player games to be solved analytically to date.

As described in the beginning of this section, each player's strategy stores the logits for a softmax parameterization of an action distribution at each information set. 2-player Kuhn poker has 6 information sets per player with 2 actions each, yielding $6 \times (2 - 1) = 6$ parameters per player and $6 \times 2 = 12$ parameters in total. 3-player Kuhn poker has 16 information sets per player with 2 actions each, yielding $16 \times (2 - 1) = 16$ parameters per player and $16 \times 3 = 48$ parameters in total. The utility function for each player is the expected payoff of the resulting strategy profile.

In this experiment, we also test *low-rank* affine best-response functions of the form $b_\theta(x) = (I + UV)x + b$ where $r \ll n$ is a *rank*, $V \in \mathbb{R}^{r \times n}, U \in \mathbb{R}^{n \times r}$, and $(U, V, b) = \theta$. This reduces the

|     | H    | T    |
| --- | ---- | ---- |
| H   | 1, 0 | 0, 1 |
| T   | 0, 1 | 1, 0 |

(a) Pennies game (2 players).

|     | R     | P     | S     |
| --- | ----- | ----- | ----- |
| R   | 0, 0  | −1, 1 | 1, −1 |
| P   | 1, −1 | 0, 0  | −1, 1 |
| S   | −1, 1 | 1, −1 | 0, 0  |

(b) Rock paper scissors game.

| 1, 0 | 0, 1 | 0, 0 |
| ---- | ---- | ---- |
| 0, 0 | 1, 0 | 0, 1 |
| 0, 1 | 0, 0 | 1, 0 |

(c) Shapley game.

Table 1: Normal-form games.

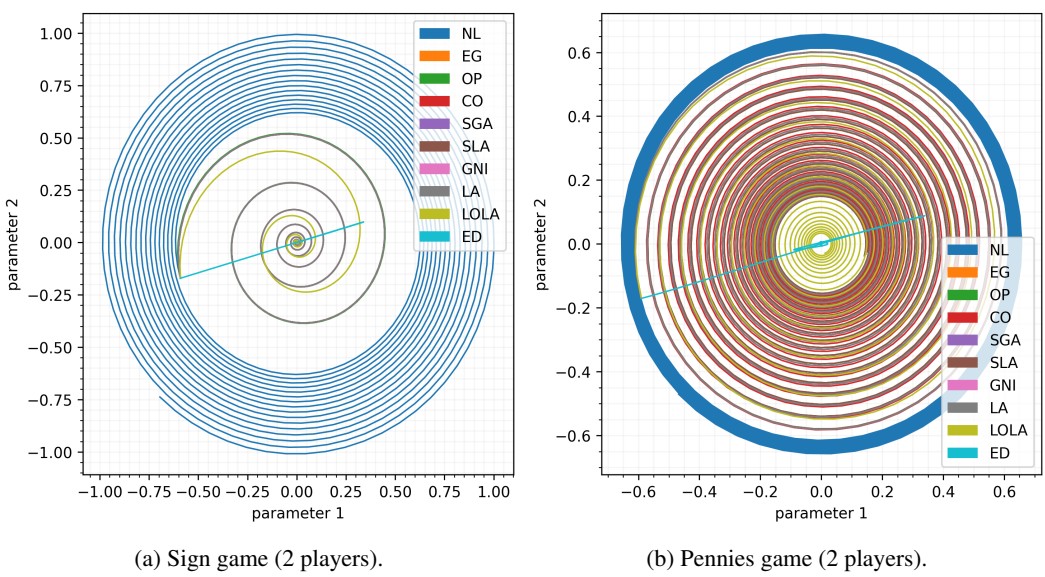

(a) Sign game (2 players).      (b) Pennies game (2 players).

Figure 1: Trajectories of each player's parameter.

number of parameters of the linear map from $n^2$ to $2nr$ and its computational complexity from $\Theta(n^2)$ multiplications and additions to $\Theta(nr)$ multiplications and additions.[1]

Figures 2k and 2l illustrate the performance of each algorithm on the 2-player and 3-player game, respectively, with $\gamma = 1$. Our algorithm significantly outperforms the rest and attains lower exploitability. The low-rank versions are suffixed with the rank number. Interestingly, some of them match the performance of the full-rank version, at a reduced computation cost.

## 5 CONCLUSIONS AND FUTURE RESEARCH

We studied the problem of computing approximate Nash equilibria of games with continuous strategy sets. Many settings involving space, time, money, and other resources are naturally modeled as continuous games. The main measure of closeness to Nash equilibrium is *exploitability*, which measures how much players can benefit from unilaterally changing their strategy. We proposed a new method that minimizes an approximation of the exploitability with respect to the strategy profile. This approximation is computed using *learned best-response functions*, which take the current strategy profile as input and return learned best responses. The strategy profile and best-response functions are trained simultaneously, with the former trying to minimize exploitability and the latter trying to maximize it. We evaluated our method in various continuous games, showing that it outperforms prior methods. We now discuss some possible extensions of our method to more general settings and directions for future research.

---

[1]To handle very large strategy vectors in very large games, one could also use sparse matrices or *hierarchical matrices* (Börm et al., 2003; Grasedyck & Hackbusch, 2003; Bebendorf, 2008; Hackbusch, 2015). The latter rely on local low-rank approximations and are closely related to the fast multipole method (Greengard & Rokhlin, 1987; Greengard, 1987; White & Head-Gordon, 1994), which is widely used in computational science to speed up calculations.

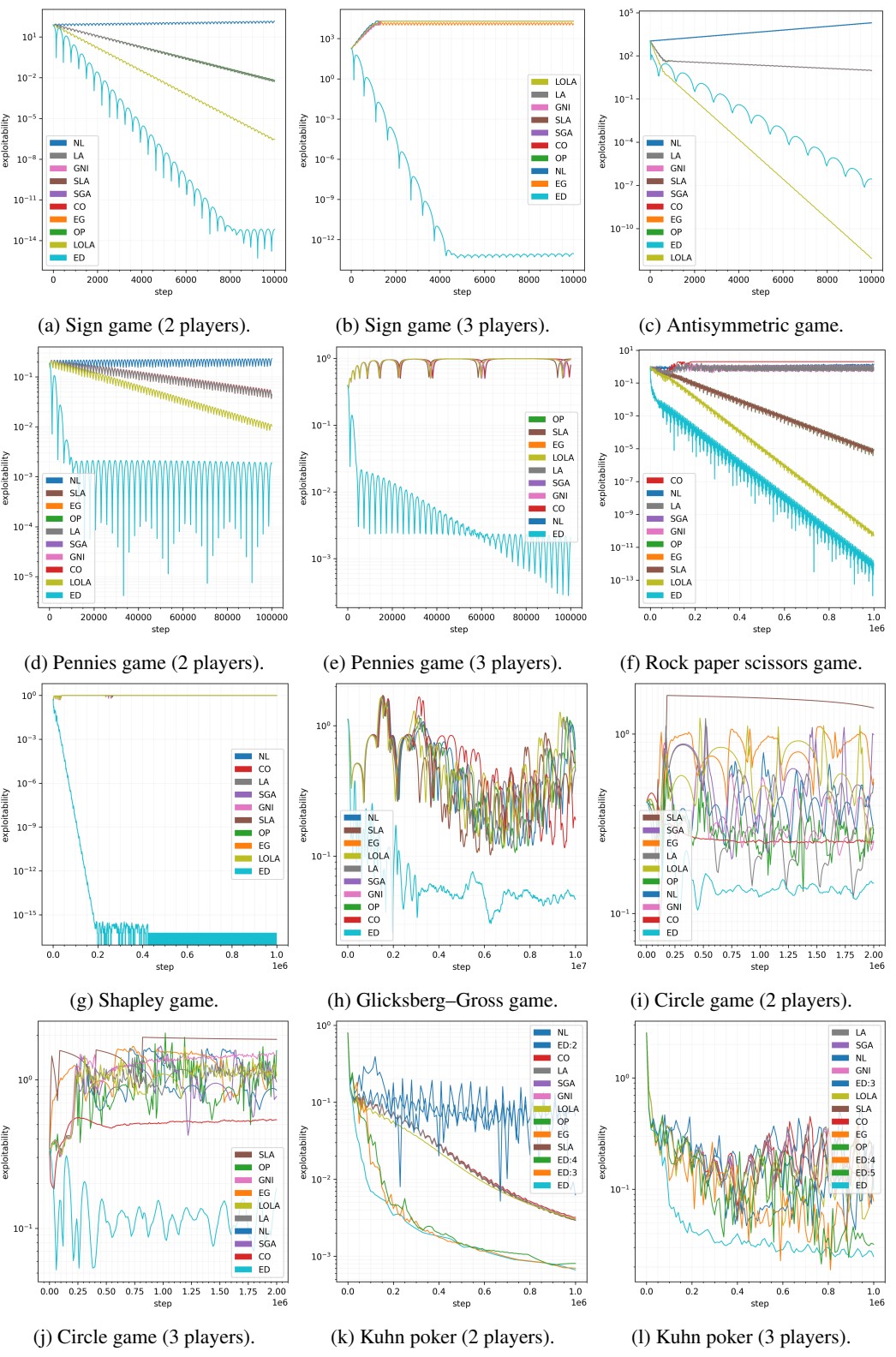

(a) Sign game (2 players).

(b) Sign game (3 players).

(c) Antisymmetric game.

(d) Pennies game (2 players).

(e) Pennies game (3 players).

(f) Rock paper scissors game.

(g) Shapley game.

(h) Glicksberg–Gross game.

(i) Circle game (2 players).

(j) Circle game (3 players).

(k) Kuhn poker (2 players).

(l) Kuhn poker (3 players).

Figure 2: Exploitabilities.

**Deep generative models**   In some games, simple parametric distributions (including mixture models) may not be flexible enough to represent equilibrium strategies well. Instead, one can use *deep generative models*. One example is a *generative network*, which transforms input noise from some fixed latent distribution (such as a multivariate Gaussian) and yields a new output distribution. This is used in GANs. Since the action space distribution is no longer a simple parametric distribution, we cannot feed those parameters into the best-response function. Instead, we can feed the generative network's weights, since they implicitly define the distribution. If the generative network is complex, it might be difficult for the best-response function to "make sense of" the weights it receives as input. In that case, we can take a more black-box approach: We do not care about the *internals* of the generative network (its intrinsic behavior), only the distribution it *produces* (its extrinsic behavior) on the action space. Instead of feeding the parameters of the generative network, we can feed a *batch of samples* from the generative network, letting the best-response function "analyze" said batch in some fashion. If the batch size is large enough, the best-response function could identify the features of the distribution that are relevant to recommending a good best response. We describe a few approaches for "analyzing" the distribution from batches of samples.

**Packing**   Lin et al. (2020) proposed a way to handle the mode collapse problem for GANs called *packing*. It modifies the discriminator to make decisions based on *multiple* samples from the same class, either real or artificially generated. The authors use analysis tools from binary hypothesis testing, particularly the seminal result of Blackwell (1953), to prove a fundamental connection between packing and mode collapse. Specifically, they show that if the discriminator is allowed to see samples from the $m$-th order *product distributions* $P^m$ and $Q^m$ instead of the usual target distribution $P$ and generator distribution $Q$, then the corresponding loss when training the generator naturally penalizes generator distributions with strong mode collapse. Based on these insights, they propose *PacGAN*, which passes $m$ "packed" or concatenated samples to the discriminator that are jointly classified as either real or artificially generated.

**Generalized moments**   An alternative approach is to use a generalized version of the *method of moments* from statistics. We let $b_\theta(x) = g_\theta(\mathrm{E}_{a\sim x} f_\theta(a))$ where $f_\theta : \prod_i A_i \to \mathbb{R}^n$ and $g_\theta : \mathbb{R}^n \to \prod_i X_i$ are trainable neural networks. Ordinary moments can be recovered with appropriate functions. For example, the first and second moments $\mathrm{E}[X]$ and $\mathrm{E}[X^2]$ are sufficient to recover the concentration parameters of a beta distribution from its variates $X$. To estimate such generalized moments, we can replace the expectation with an average $\frac{1}{|J|} \sum_{j\in J} f_\theta(a_j)$, where the $a_j$ are independent samples of action profiles from the strategy profile.

**Multi-state games**   The above can be readily generalized to perfect-information games where all players receive a common observation that is correlated with the payoffs given their actions. In that case, the best-response function can feed the observation through the strategy networks they take as input and analyze the resulting action distributions.

The case of imperfect-information games is more complex. In that case, best responses do not have access to the other players' observations, which their action distributions depend on. In that case, we have at least two options. On the one hand, the best-response functions could take the strategy networks' parameters as input (the intrinsic approach), since these implicitly define all of their behavior. On the other hand, they could analyze the strategies purely through their output behavior (the extrinsic approach). In that case, they would have to *learn to query* such networks with hypothetical observations, given their own observation.

When designing the (parameterized) best-response function, one could also exploit the special structure of the extensive-form game by performing some form of *local game tree analysis* of the strategy profile in, say, the neighborhood of an information set. See Timbers et al. (2022) for an application of this in the extensive-form setting that computes approximate best responses.

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

## A ADDITIONAL RELATED RESEARCH

McMahan et al. (2003) introduced the double oracle algorithm for normal-form games and proved its convergence. Adam et al. (2021) extended it to two-player zero-sum continuous games. Kroupa & Votroubek (2021) extended it to $n$-player continuous games. Their algorithm maintains finite strategy sets for each player and iteratively extends them with best responses to an equilibrium of the induced finite sub-game. This "converges fast when the dimension of strategy spaces is small, and the generated subgames are not large." For example, in the two-player zero-sum case: "The best responses were computed by selecting the best point of a uniform discretization for the one-dimensional problems and by using a mixed-integer linear programming reformulation for the Colonel Blotto games."

Ganzfried (2021) introduced an algorithm for approximating equilibria in continuous games called "redundant fictitious play" and applied it to a continuous Colonel Blotto game. Kamra et al. (2019) presented DeepFP, an approximate extension of fictitious play (Brown, 1951; Berger, 2007) to continuous action spaces. They demonstrate stable convergence to equilibrium on several classic games and a large forest security domain. DeepFP represents players' approximate best responses via generative neural networks, which are highly expressive implicit density approximators. The authors state that, because implicit density models cannot be trained directly, they employ a game-model network that is a differentiable approximation of the players' payoffs given their actions, and train these networks end-to-end in a model-based learning regime. This allows working in the absence of gradients for players.

Li & Wellman (2021) extended the double oracle approach to $n$-player general-sum continuous Bayesian games. They represent agents as neural networks and optimize them using *natural evolution strategies (NES)* (Wierstra et al., 2008; 2014). For pure equilibrium computation, they formulate the problem as a bi-level optimization and employ NES to implement both inner-loop best-response optimization and outer-loop regret minimization. Bichler et al. (2021) presented a learning method that represents strategies as neural networks and applies simultaneous gradient dynamics to provably learn local equilibria. Fichtl et al. (2022) compute distributional strategies on a discretized version of the game via online convex optimization, specifically *simultaneous online dual averaging (SODA)*, and show that the equilibrium of the discretized game approximates an equilibrium in the continuous game.

In a *generative adversarial network (GAN)* (Goodfellow et al., 2014), a generator learns to generate fake data while a discriminator learns to distinguish it from real data. Metz et al. (2016) introduced a method to stabilize GANs by defining the generator objective with respect to an unrolled optimization of the discriminator. They show how this technique solves the common problem of mode collapse, stabilizes training of GANs with complex recurrent generators, and increases diversity and coverage of the data distribution by the generator. Grnarova et al. (2019) proposed using an approximation of the game-theoretic *duality gap* as a performance measure for GANs. Grnarova et al. (2021) proposed using this measure as the objective, proving some convergence guarantees.

Lockhart et al. (2019) presented *exploitability descent*, which computes approximate equilibria in two-player zero-sum extensive-form games by direct strategy optimization against worst-case opponents. They prove that the exploitability of a player's strategy converges asymptotically to zero. Hence, when both players employ this optimization, the strategy profile converges to an equilibrium. Unlike extensive-form fictitious play (Heinrich et al., 2015) and counterfactual regret minimization (Zinkevich et al., 2007), their convergence pertains to the strategies being optimized rather than the time-average strategies. Timbers et al. (2022) introduced approximate exploitability, which uses an approximate best response computed through search and reinforcement learning. This is a generalization of *local best response*, a domain-specific evaluation metric used in poker (Lisý & Bowling, 2017).

Fiez et al. (2022) consider minimax optimization $\min_x \max_y f(x, y)$ in the context of two-player zero-sum games, where the min-player (controlling $x$) tries to minimize $f$ assuming the max-player (controlling $y$) then tries to maximize it. In their framework, the min-player plays against *smooth algorithms* deployed by the max-player (instead of full maximization, which is generally NP-hard). Their algorithm is guaranteed to make monotonic progress, avoiding limit cycles or diverging behavior, and finds an appropriate "stationary point" in a polynomial number of iterations. This work has important differences to ours. First, our work tackles multi-player general-sum games, a more general class of games than two-player zero-sum games. Second, their work does not use learned best-response functions, but instead runs a multi-step optimization procedure for the opponent on every iteration, with the opponent parameters re-initialized from scratch. This can be expensive for complex games, which may require many iterations to learn a good opponent strategy. It also does not reuse information from previous iterations to recommend a good response. Our learned best-response functions can retain information from previous iterations and do not require a potentially expensive optimization procedure on each iteration.

Gemp et al. (2022) proposed an approach called *average deviation incentive descent with adaptive sampling* that iteratively improves an approximation to a Nash equilibrium through joint play by tracing a homotopy that defines a continuum of equilibria for the game regularized with decaying

levels of entropy. To encourage iterates to remain near this path, they minimize average deviation incentive via stochastic gradient descent.

Ganzfried & Sandholm (2010a;b) presented a procedure for solving large imperfect information games by solving an infinite approximation of the original game and mapping the equilibrium to a strategy profile in the original game. Counterintuitively, it is often the case that the infinite approximation can be solved much more easily than the finite game. The algorithm exploits some qualitative model of equilibrium structure as an additional input in order to find an equilibrium in continuous games.

Mazumdar et al. (2019) proposed local symplectic surgery, a two-timescale procedure for finding local Nash equilibria in two-player zero-sum games. They showed that previous gradient-based algorithms cannot guarantee convergence to local Nash equilibria due to the existence of non-Nash stationary points. By taking advantage of the differential structure of the game, they construct an algorithm for which the local Nash equilibria are the only attracting fixed points. Mazumdar et al. (2020) analyze the limiting behavior of competitive gradient-based learning algorithms using dynamical systems theory. They characterize a non-negligible subset of the local Nash equilibria that will be avoided if each agent employs a gradient-based learning algorithm.

Mertikopoulos & Zhou (2019) examines the convergence of no-regret learning in games with continuous action sets, focusing on learning via "dual averaging", a widely used class of no-regret learning schemes where players take small steps along their individual payoff gradients and then "mirror" the output back to their action sets. They introduce the notion of variational stability, and show that stable equilibria are locally attracting with high probability whereas globally stable equilibria are globally attracting with probability 1.

Willi et al. (2022) show that the original formulation of the LOLA method (and follow-up work) is inconsistent in that it models other agents as naive learners rather than LOLA agents. In previous work, this inconsistency was suggested as a cause of LOLA's failure to preserve stable fixed points (SFPs). They formalize consistency and show that *higher-order LOLA (HOLA)* solves LOLA's inconsistency problem if it converges. They also proposed a new method called *consistent LOLA (COLA)*, which learns update functions that are consistent under mutual opponent shaping. It requires no more than second-order derivatives and learns consistent update functions even when HOLA fails to converge.

Perolat et al. (2022) introduce *DeepNash*, an autonomous agent capable of learning to play the imperfect information game Stratego from scratch, up to a human expert level. DeepNash uses a game-theoretic, model-free deep reinforcement learning method, without search, that learns to master Stratego via self-play. The *Regularised Nash Dynamics (R-NaD)* algorithm, a key component of DeepNash, converges to an approximate Nash equilibrium, instead of "cycling" around it, by directly modifying the underlying multiagent learning dynamics. Qin et al. (2022) proposed a no-regret style reinforcement learning algorithm *PORL* for continuous action tasks, proving that it has a last-iterate convergence guarantee.

## B    ADDITIONAL FIGURES

In this section, we present some additional figures. Figure 3 illustrates the probability density functions of each player after training in the Glicksberg–Gross game and the circle game with 3 players. The black dashed line indicates the analytic equilibrium. Figure 4 illustrates the trajectories of each player's action probabilities in the rock paper scissors game and the Shapley game. Since there are 3 actions, the action probabilities are points on a 2-simplex. The remaining figures compare different values of the hyperparameters $\gamma$ and $\eta$ (the learning rate). They yielded qualitatively similar results.

## C    CODE SAMPLE

Listing 1 shows an example implementation of our method in the Python programming language using Google JAX (Bradbury et al., 2018). We used Python version 3.10.7 and JAX version 0.3.24 for our experiments.

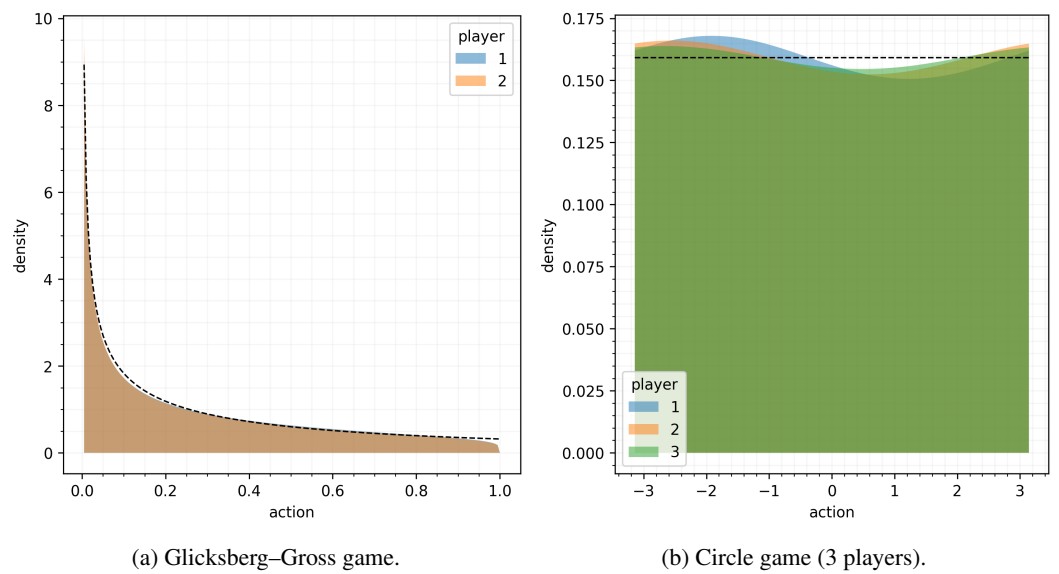

(a) Glicksberg–Gross game.

(b) Circle game (3 players).

Figure 3: Probability density functions after training.

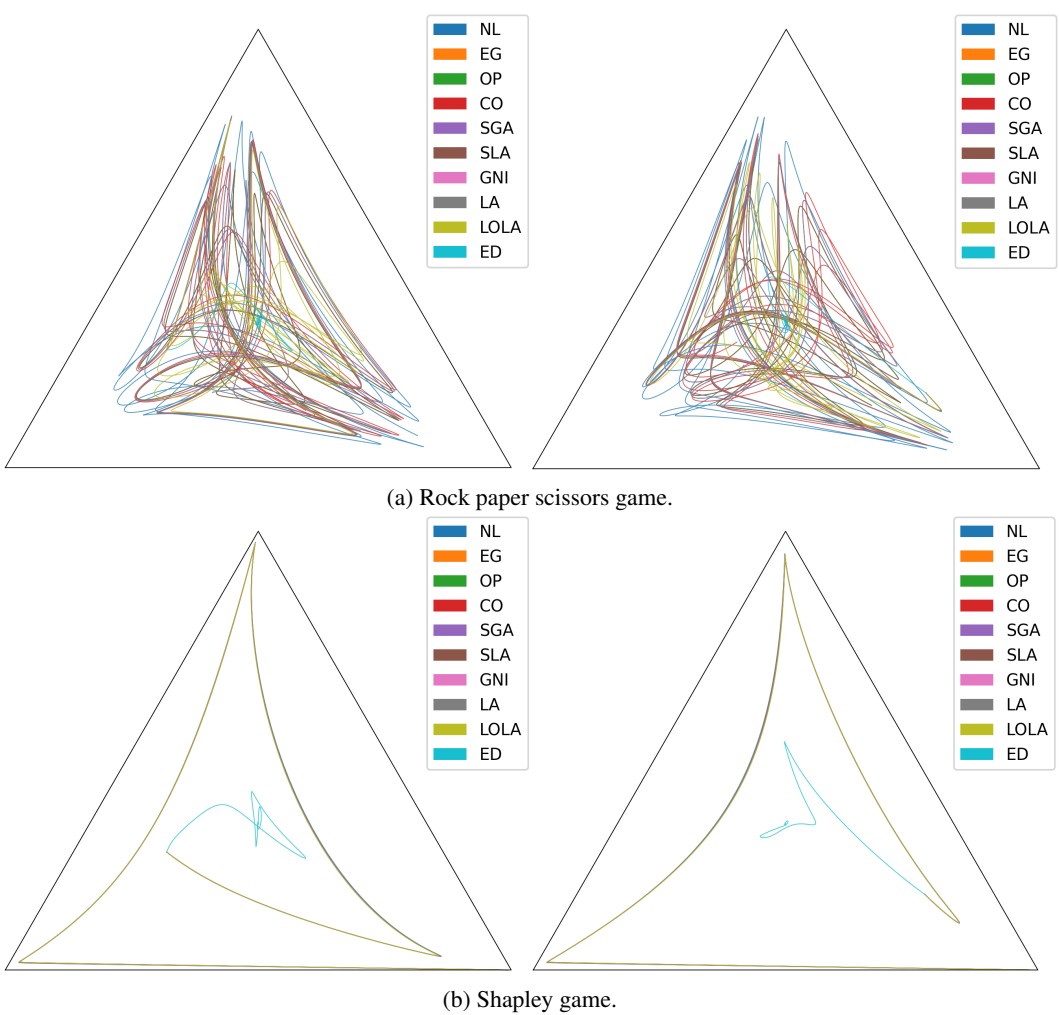

(a) Rock paper scissors game.

(b) Shapley game.

Figure 4: Trajectories of each player's action probabilities. Left: Player 1. Right: Player 2.

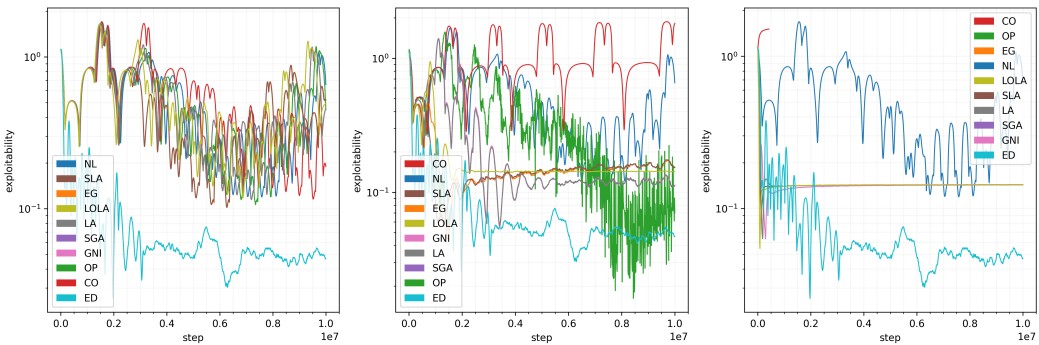

Figure 5: Glicksberg–Gross game with $\gamma = 0.1, 1, 10$ (left to right).

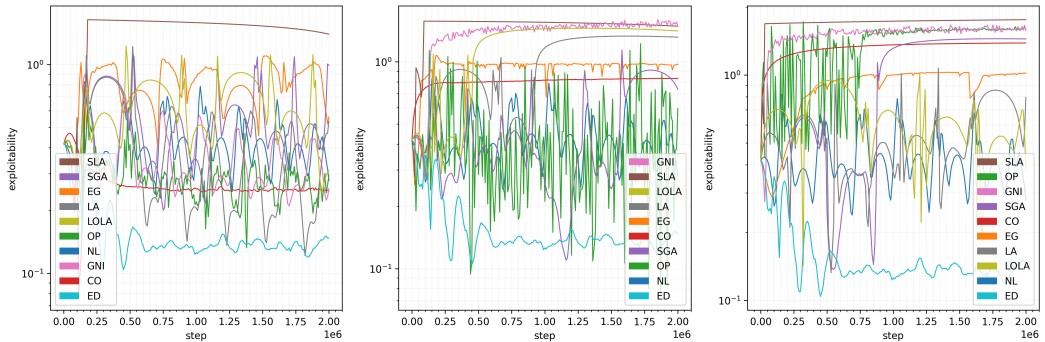

Figure 6: 2-player circle game with $\gamma = 0.1, 1, 10$ (left to right).

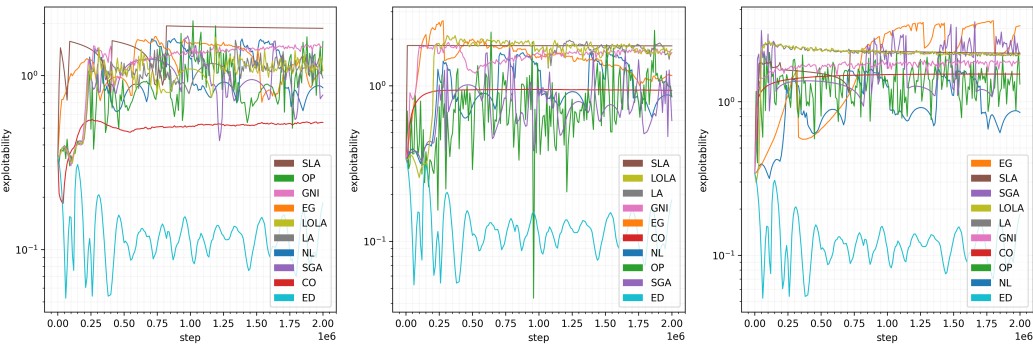

Figure 7: 3-player circle game with $\gamma = 0.1, 1, 10$ (left to right).

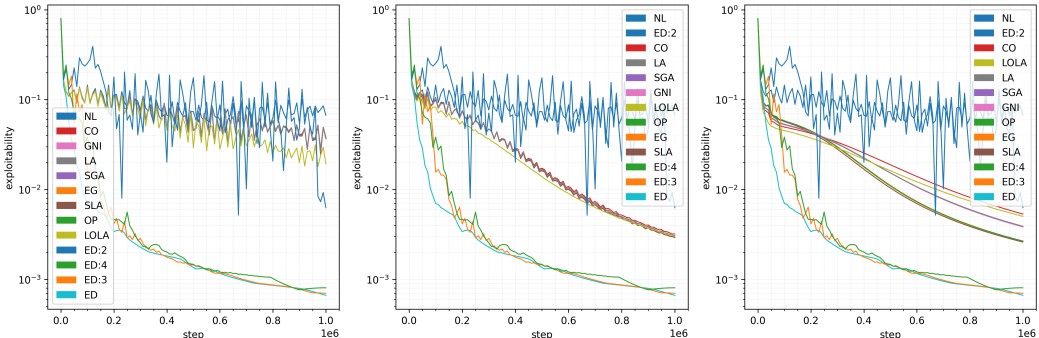

Figure 8: 2-player Kuhn poker with $\gamma = 0.1, 1, 10$ (left to right).

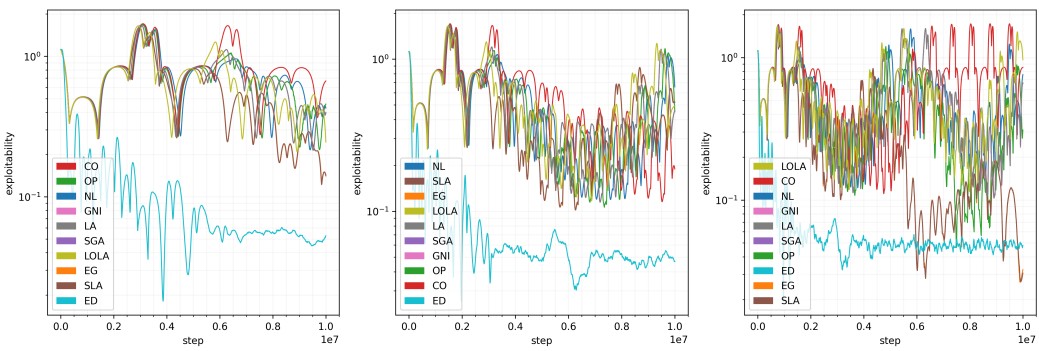

Figure 9: Glicksberg–Gross game with $\eta = 5 \times 10^{-5}, 10^{-4}, 2 \times 10^{-4}$ (left to right).

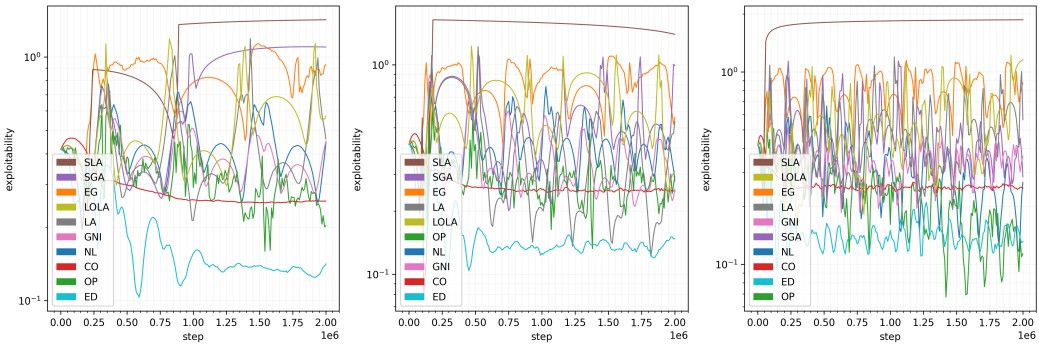

Figure 10: 2-player circle game with $\eta = 5 \times 10^{-5}, 10^{-4}, 2 \times 10^{-4}$ (left to right).

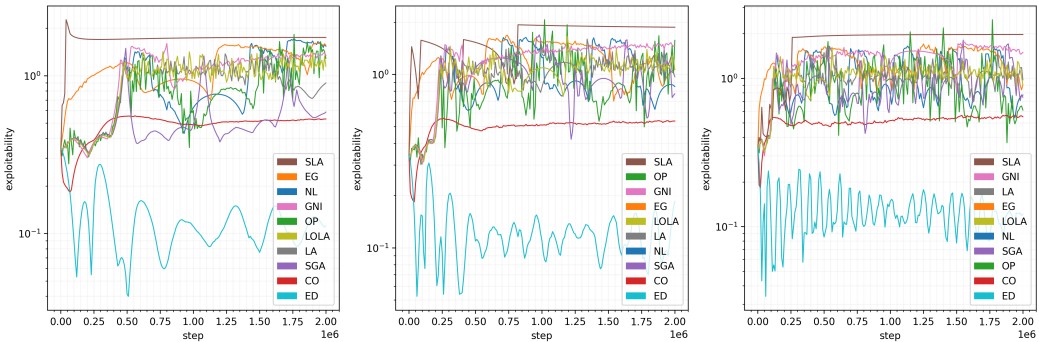

Figure 11: 3-player circle game with $\eta = 5 \times 10^{-5}, 10^{-4}, 2 \times 10^{-4}$ (left to right).

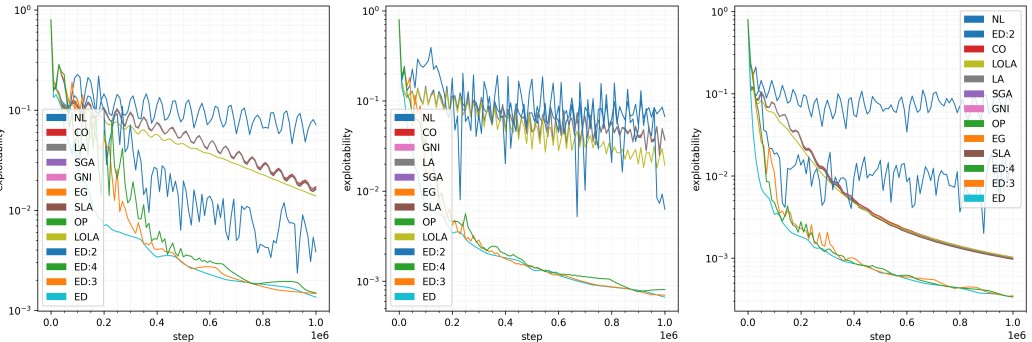

Figure 12: 2-player Kuhn poker with $\eta = 5 \times 10^{-3}, 10^{-2}, 2 \times 10^{-2}$ (left to right).

**Listing 1** Example implementation of our method.

```python
from jax import grad, tree_map
from jax.lax import fori_loop
from jax.flatten_util import ravel_pytree

def replace(lst, index, value):
    return lst[:index] + [value] + lst[index + 1:]

def nikaido_isoda(u, x, y):
    return sum(
        u(replace(x, i, yi))[i]
        for i, yi in enumerate(y)
    ) - sum(u(x))

def best_response_fn(x, w):
    x_vector, unravel = ravel_pytree(x) # ravel x into a flat vector
    A, b = w
    y_vector = A @ x_vector + b
    return unravel(y_vector) # unravel y from a flat vector

def run(u, x, w, n: int, lr: float):
    '''
    u: utility function
    x: strategy profile
    w: best-response function parameters
    n: number of steps
    lr: learning rate
    The strategy profile should be a list of pytrees. The utility function should take a
    ↪   strategy profile as input and return a sequence of the same length containing each
    ↪   player's utility.
    '''

    def objective(x, w):
        y = best_response_fn(x, w) # best responses
        return nikaido_isoda(u, x, y)

    def update(step, state):
        x, w = state
        dx, dw = grad(objective, [0, 1])(x, w)
        x_new = tree_map(lambda x, dx: x - dx * lr, x, dx)
        w_new = tree_map(lambda w, dw: w + dw * lr, w, dw)
        return x_new, w_new

    x_final, w_final = fori_loop(0, n, update, (x, w))
    return x_final
```

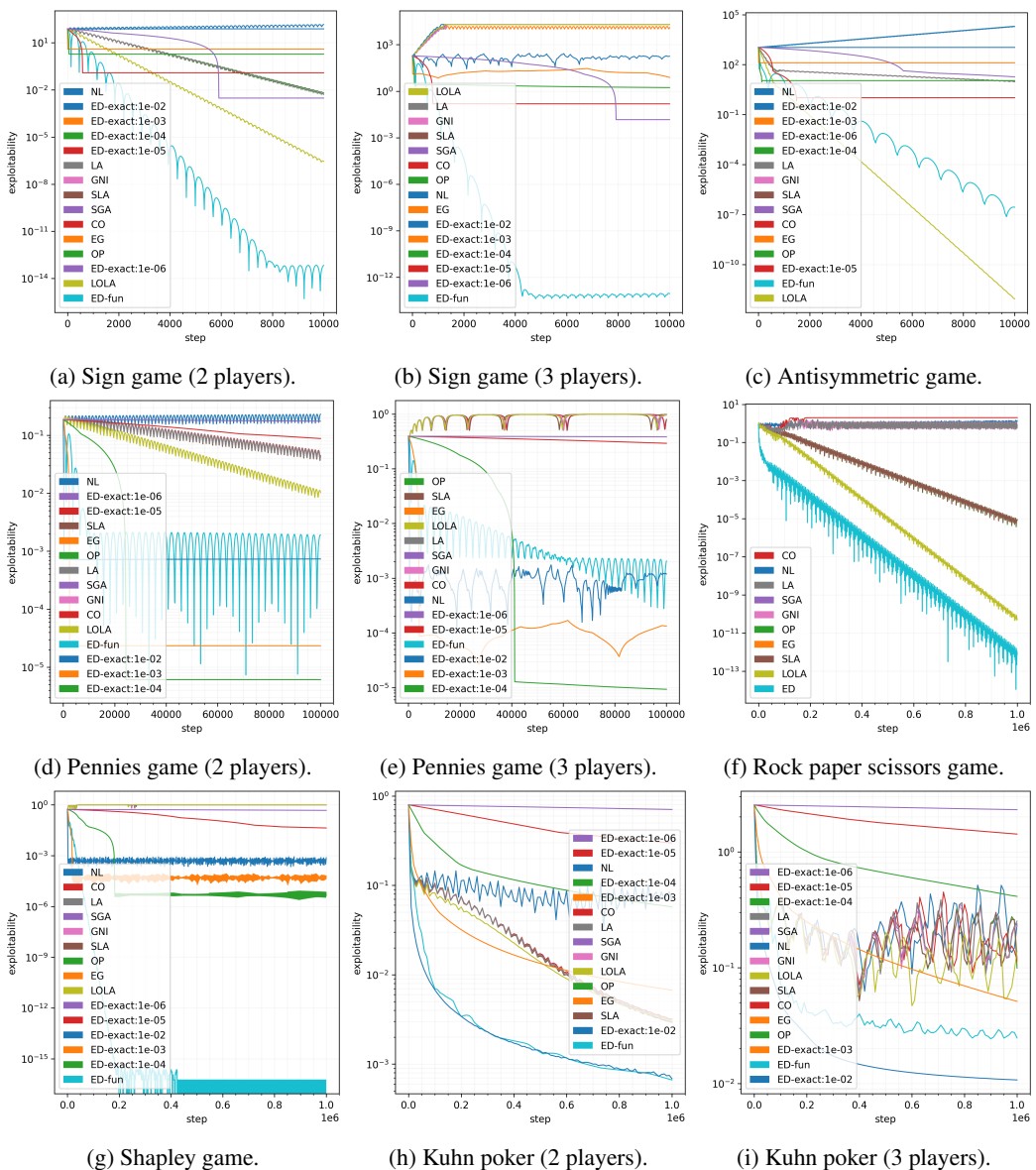

(a) Sign game (2 players).

(b) Sign game (3 players).

(c) Antisymmetric game.

(d) Pennies game (2 players).

(e) Pennies game (3 players).

(f) Rock paper scissors game.

(g) Shapley game.

(h) Kuhn poker (2 players).

(i) Kuhn poker (3 players).

Figure 13: Exploitabilities.

# D    COMPARISON TO EXACT EXPLOITABILITY DESCENT

In many situations, we do not have access to exact best responses. Computing them can be inefficient or intractable in complex games. Our method does not require them, and thus avoids that bottleneck. If we do have access to exact best-response oracles, we can try performing gradient descent on the *exact* exploitability function $\psi(x) = \sup_y \phi(x, y)$, as in Lockhart et al. (2019). (Because of the supremum, $\psi$ may not be differentiable even if $\phi$ is.)

Figure 13 compares our method (labeled ED-fun) to exact exploitability descent (labeled ED-exact:$\eta$ where $\eta$ is a learning rate) on some games where we have access to the exact exploitability function (more precisely, its subgradients). On some of those games, ED-exact decreases quickly at first, but stops at a higher exploitability than ED-fun. This is likely because the exact exploitability function $\sup_y \phi(x, y)$ may fail to be differentiable when the approximate exploitability function $\phi(x, b_\theta(x))$ is. In that case, subgradient methods might be useful.

