# OpenReview forum: "Equilibrium-finding via exploitability descent with learned best-response functions"
_ICLR.cc/2023/Conference — Submitted to ICLR 2023_

### Official Review · Reviewer_WPMd · 2022-10-24

**Confidence:** 4
**Correctness:** 3
**Technical Novelty And Significance:** 2
**Empirical Novelty And Significance:** 2
**Recommendation:** 5

**Clarity, Quality, Novelty And Reproducibility:**

The paper presents the idea clearly and the idea of using a learned response function to compute Nash equlibria is also interesting.
The experimental results are extensive but it is very challenging to reproduce these results without open-sourced codes.


**Strength And Weaknesses:**

Strength:
1. The authors did a thorough literature review on Nash equilibrium for continuous games and also compared the proposed method with these existing ones.
2. The authors experimented with the proposed method on variant games from a naive sign game to Kuhn poker.

Weaknesses:
1. The authors assume that there exists an explicit fully-fledged function mapping different strategies for Player 1 to different strategies for Player 2. This is a very strong assumption and not well justified. In the experiment, the authors used a linear form best-response function which is an even stronger assumption. What if there are multiplier players and there are interactions among multiple players? Will the linearity still hold?
2. The learning rate is fixed at 0.01 for all experiments except for Glicksberg–Gross game 10-4. It will be helpful to have some ablation study w.r.t. this hyperparameter if it needs to be carefully set. The same issue for the hyperparameter gamma. It is worth exploring how these two hyperparameters affect the convergence jointly.
3. The authors claimed that 'If X is infinite and Y is nontrivial, X → Y is infinite-dimensional' and proposed neural networks to model the function b. This is an important use case but it seems that the authors did not offer any experimental results on it.
4. In Fig. 1(a), it is not clear how the parameters of OP, SGA, CO, EG, and GNI evolve. It will help visualize the results to use different marks for different methods. The same issue for Figure 2.
5. In Fig. 2(l), why ED(4) with rank 4 is better than the full-rank affine best response function? It is worth doing more ablation study w.r.t. the low rank when decomposing matrix A into I+UV.


**Summary Of The Paper:**

The authors propose a new method to compute approximate Nash equilibria of games with continuous strategy sets. Specifically, the new method minimizes an approximation of exploitability with respect to the strategy profile using learned best-response functions.

**Summary Of The Review:**

Considering the paper's strong assumptions and their justifications (more details in Weaknesses), I would not recommend it for publication in its present form.

I have read the authors' response and would like to keep the same score.

---

> ### Author Response · Authors · 2022-11-15
> **Response**
>
> We thank the reviewer for their feedback.
>
> "The authors assume that there exists an explicit fully-fledged function mapping different strategies for Player 1 to different strategies for Player 2."
>
> Since our method applies to multi-player games beyond two players, we assume the reviewer is referring to the single-player perspective in a zero-sum game, which we use to motivate our method.
>
> To clarify a possible misunderstanding, we do not assume that any such best-response function is given to us, explicitly or otherwise. Rather, it is entirely up to us to learn how to map Player 1 strategies to *approximate* best responses.
>
> Is the question under what mathematical conditions it is true that, for any Player 1 strategy, there exists a Player 2 strategy that exactly minimizes Player 1's payoff? Or is the comment about a function being given, which is not the case in our paper?
>
> "the authors used a linear form best-response function which is an even stronger assumption"
>
> Yes, an affine best response function cannot capture exact best responses for general games. But we wanted to introduce our method in its simplest form and show that, even with this simple best-response model class, it already outperforms prior methods. As stated in the paper, it is also possible for the best response function to take on a more complex form, such as a neural network.
>
> "What if there are multiplier players and there are interactions among multiple players? Will the linearity still hold?"
>
> Even in the zero-sum setting, there are games whose best response functions are not affine, but see our comments above. Empirically, our experiments suggest that the affine best response function also works well for multiple players. This might be due to the fact that a best response function that is "locally accurate" is sufficient to yield the desired dynamics (of convergence to a local Nash equilibrium).
>
> "It is worth exploring how these two hyperparameters affect the convergence jointly."
>
> We have added experiments comparing different values of these hyperparameters to the appendix. They yielded qualitatively similar results.
>
> "This is an important use case but it seems that the authors did not offer any experimental results on it."
>
> On the games we tested, affine best response functions were sufficient. However, there might be other games that would benefit from using a more complex best response function. In any case, as noted in that paragraph, neural networks are also technically finite-dimensional, since they have a finite number of real parameters. We have to approximate the infinite-dimensional function space with a finite-dimensional parameterization.
>
> "In Fig. 1(a), it is not clear how the parameters of OP, SGA, CO, EG, and GNI evolve."
>
> This figure is described in the experiments section under "Sign game", which states that the trajectories of all methods from OP to LA (inclusive) overlap. By this, we mean that they overlap exactly. We can clarify this.
>
> "It will help visualize the results to use different marks for different methods. The same issue for Figure 2."
>
> In Figure 2, many of the trajectories overlap almost exactly, so adding different marks would not help clarity. Instead, we have added a more detailed textual description.
>
> "In Fig. 2(l), why ED(4) with rank 4 is better than the full-rank affine best response function?"
>
> Their final exploitability is very similar (note the logarithmic scale). We believe the difference is probably due to noise.
>
> "It is worth doing more ablation study w.r.t. the low rank when decomposing matrix A into I+UV."
>
> Does the reviewer have a particular set of ranks in mind, beyond those we tested?
>
> "The experimental results are extensive but it is very challenging to reproduce these results without open-sourced codes."
>
> Our full codebase for this paper overlaps with other papers. We intend to disentangle and release it as soon as possible. In the meantime, we have added some code to the appendix. If the reviewer would like to see other specific pieces of code, please let us know.

---

> > ### Comment · Reviewer_WPMd · 2022-11-28
> > **Keep the same score**
> >
> > Thank you for your response.
> > 1. I am not still convinced by the authors' explanations of explicit fully-fledged functions. I was asking 1) how real this assumption is and 2) how feasible the proposed method is able to learn it. It is also not fair to claim a neural network and experiment with a linear one.
> > 2. For the low-rank decomposition, it is worth exploring the performance by varying the low dimensionality r= 2,3,4,5 at least.

---

> > > ### Author Response · Authors · 2022-12-04
> > > **Comment**
> > >
> > > Thank you for your feedback.
> > >
> > > 1.1: "I was asking 1) how real this assumption is"
> > >
> > > Suppose $Y$ is compact and $u : X \times Y \to \mathbb{R}$ is continuous in its second argument. Let $x \in X$. By the extreme value theorem, a continuous real-valued function on a non-empty compact set attains its extrema. Therefore, there exists $y \in Y$ such that $u(x, y) = \sup_{y \in Y} u(x, y)$. Since this is true for every $x \in X$, there exists a function $b : X \to Y$ such that, for every $x \in X$, $u(x, b(x)) = \sup_{y \in Y} u(x, y)$. That is, $b$ is a best-response function.
> > >
> > > Even when $Y$ is not compact and $u$ does not attain its extrema, one can define a best-response *value* for any $x \in X$ as $\sup_{y \in X} u(x, y)$, provided the latter exists. In that case, we have the following: Let $\varepsilon > 0$. Let $x \in X$. Any function gets arbitrarily close to its supremum (continuity is not required). Therefore, there exists a $y \in Y$ such that $u(x, y) + \varepsilon \geq \sup_{y \in Y} u(x, y)$. Therefore, there exists a function $\tilde{b} : X \to Y$ such that, for every $x \in X$, $u(x, \tilde{b}(x)) + \varepsilon \geq \sup_{y \in Y} u(x, y)$. That is $\tilde{b}$ is an $\varepsilon$-*approximate* best-response function.
> > >
> > > We are interested in *approximate* best responses and exploitability. Therefore, for our purposes, it is sufficient that, for every $x \in X$, $\sup_{y \in Y} u(x, y)$ exists, which is a quite weak assumption.
> > >
> > > 1.2: "2) how feasible the proposed method is able to learn it."
> > >
> > > The experiments, which include several games, show that the proposed method is able to learn it.
> > >
> > > 1.3: "It is also not fair to claim a neural network and experiment with a linear one."
> > >
> > > We meant to say that one *can* use a neural network. We have amended the text accordingly.
> > >
> > > (As a parenthetical, an affine function is technically a neural network, just one with zero hidden layers.)
> > >
> > > We have also added a new section to the appendix (Appendix E) containing experiments with a neural network best-response function.
> > >
> > > 2: "For the low-rank decomposition, it is worth exploring the performance by varying the low dimensionality r= 2,3,4,5 at least."
> > >
> > > We have added these to Figure 2, in the revised PDF of the paper.

---

> > > ### Author Response · Authors · 2022-12-06
> > > **Comment**
> > >
> > > Since new revisions can no longer be uploaded, here is a summary of the results: We test our method with neural network best-response functions, including various hidden layer sizes. We use the standard He initialization to initialize the weights. While on some games their performance is comparable to that of affine best-response functions, on other games they yield significantly faster convergence. This is probably due to the fact that neural networks are a more flexible (in particular, nonlinear) class of functions. For example, we have the following exploitabilities:
> > >
> > > 2-player sign game at 2e3 steps
> > > * affine BRF: 1e-2
> > > * neural network BRF: 1e-14
> > >
> > > 3-player sign game at 2e3 steps
> > > * affine BRF: 1e-4
> > > * neural network BRF: 1e-10
> > >
> > > antisymmetric game at 6e3 steps
> > > * affine BRF: 1e-4
> > > * neural network BRF: 1e-13
> > >
> > > 2-player pennies game at 2e4 steps
> > > * affine BRF: 1e-3
> > > * neural network BRF: 1e-15
> > >
> > > 3-player pennies game at 2e4 steps
> > > * affine BRF: 1e-2
> > > * neural network BRF: 1e-15
> > >
> > > rock paper scissors at 1e5 steps
> > > * affine BRF: 1e-3
> > > * neural network BRF: 1e-15

---

### Official Review · Reviewer_KfZH · 2022-10-24

**Confidence:** 4
**Correctness:** 3
**Technical Novelty And Significance:** 3
**Empirical Novelty And Significance:** 3
**Recommendation:** 6

**Clarity, Quality, Novelty And Reproducibility:**

The paper is well-organized and easy to follow. The designed algorithm is basically a proper combination of exploitability descent (one previous work) and the idea of lookahead (a series of previous work) through learned best-response functions, which themselves are not novel independently. The experiments are clearly described with high reproducibility.

**Strength And Weaknesses:**

Strength:
1. The iead of introducing learned best-response function is interesting. It is not surprising that replacing the observed opponents' strategy by the learned opponents' best-response will lead to better performance. On the other hand, it seems to be a simpler and more general way (like a blackbox) to doing lookahead than previous approaches by modeling opponents.
2. Plenty of experiments are conducted to evaluate the proposed algorithms. The solved games in this paper covers many common classes.

Weaknesses:
1. There is no theoretical guarantee on convergence or approximation bound.
2. The "exploitability" itself may be too simple. Anyway, there should be a comparison with simply exploitability-descent (Lockhart et al. (2019)). Some intuition is that it is the "learned best-response" part make more contributions. Thus, it would also be better if an approach like GNI + learned best response is further compared.

**Summary Of The Paper:**

The paper studies the problem computing NE for continous games. It equipes exploitability descent with learned best-response functions to provide better convergence (in empirical). Plenty of experiments are conducted to show its better performance than previous works.

**Summary Of The Review:**

The paper proposed a new algorithm to computing NE for continous games, which combines previous approaches properly. Plenty of experiments support its performance. Thus I am tend to accept it.

---

> ### Author Response · Authors · 2022-11-15
> **Response**
>
> We thank the reviewer for their feedback.
>
> "There is no theoretical guarantee on convergence or approximation bound."
>
> Theoretical guarantees are beyond the scope of this paper, though a potentially interesting question for future research. It is not uncommon in this field for methods to be introduced before theoretical guarantees are obtained. Indeed, the latter may be difficult enough to stand alone as a research contribution.
>
> "The "exploitability" itself may be too simple."
>
> We are not exactly sure what is meant by this. Could the reviewer please clarify?
>
> "there should be a comparison with simply exploitability-descent (Lockhart et al. (2019)"
>
> We can add exploitability descent with exact best-response oracles, if they are available, to the experiments. However, as noted in the paper, part of the motivation behind our method is that computing exact best responses can be intractable in complex games, and our method avoids that bottleneck. Is this what the reviewer is asking?
>
> "it would also be better if an approach like GNI + learned best response is further compared"
>
> We are not exactly sure what is meant by this. Could the reviewer please clarify? GNI uses $\nabla \phi(x, x + \gamma \xi)$ (i.e., the approximate best responses are obtained via a 1-step optimization, for each player, from the current strategy profile), whereas our method uses $\nabla \phi(x, b_\theta(x))$, where $b_\theta$ is the learned approximate best response function.

---

> > ### Comment · Reviewer_KfZH · 2022-11-15
> > **Response to Authors' response**
> >
> > Thanks for your response.
> >
> > For theoretical analysis, it is still a big issue when considering potential applications.
> >
> > I mentioned "The "exploitability" itself may be too simple." to argue that there is no contribution on proposing new metric for NE, for which I believe it is okay but I would hope for more.
> >
> > For comparison with exploitability descent with exact best-response oracles, I wonder whether the learned best response performs as well as the excat one if available.
> >
> > Never mind about the GNI issue. What I want to explore here is whether the idea of using "learned best repsonse" to overcome the battleneck of intractrable exact best response can be used in other scenario.

---

> > > ### Author Response · Authors · 2022-11-17
> > > **Comment**
> > >
> > > We have added a section to the appendix titled "Comparison to exact exploitability descent". It compares our method to exact exploitability descent in some games where the latter is available.

---

### Official Review · Reviewer_u8oJ · 2022-10-25

**Confidence:** 4
**Correctness:** 4
**Technical Novelty And Significance:** 2
**Empirical Novelty And Significance:** 2
**Recommendation:** 6

**Clarity, Quality, Novelty And Reproducibility:**

The paper is very clearly written and is of high quality.

My concerns about novelty are highlighted above.

The experimental settings are clearly described although code (or at least code samples) would help with reproducibility.

**Strength And Weaknesses:**

Regarding strengths, the paper is clearly written with the notation, motivation and the intuition of the approach being very clearly and thoroughly introduced. To the best of my knowledge, this is the first work to propose the learned best responses framework in its full generality where the learned best response function can be arbitrarily parametrized in terms of $\theta$.

Regarding weaknesses, the concept of learned best responses is not entirely new. For example, [1] proposes an instance of the learned best response framework where the adversary learns a weighted average of fixed smooth functions of the opponent's strategy. The individual algorithms may run GD on an objective for a fixed initialization for a fixed number of epochs. But in principle, any smooth function can be used. Even in this limited setting, one does not need to worry about cycling when using gradient ascent descent.

Given that learned best responses are not entirely new, I would suggest to the authors to highlight how/when the generality of their framework is useful or lessons learned about these problems from their experiments.



[1] Tanner Fiez∗, Lillian J. Ratliff, Chi Jin, Praneeth Netrapalli, MINIMAX OPTIMIZATION WITH SMOOTH ALGORITHMIC ADVERSARIES, ICLR 2022

**Summary Of The Paper:**

The authors study the problem of finding Nash equilibria in continuous games. Their key observation is that one of the reasons why algorithms for computing Nash equilibria fail is that players spend a lot of iterations re-learning the best response to a strategy of the other players. This happens even if the same strategies are repeated multiple times during the game.

The authors propose that players should learn the parameters of a best response function that learns to best respond to the other players. This function can be parametrized as affine functions or neural networks in the most general setting. Experimental results show that across a variety of games, the proposed approach can outperform prior work.

**Summary Of The Review:**

In summary, I am willing to increase my score if the authors elaborate about the uniqueness/novelty of their approach or at least highlight some lessons learned from their experiments. As it is though, I am currently leaning towards rejection.

After the responses of the authors, I have increased my score to a 6.

---

> ### Author Response · Authors · 2022-11-15
> **Response**
>
> We thank the reviewer for their feedback.
>
> First, the given reference is limited to zero-sum games.
>
> Second, they do not actually use or experiment with best response functions in a concrete sense, but instead run a multi-step optimization procedure for the opponent on every iteration, with the opponent parameters re-initialized from scratch. This can be expensive for complex games, which may require many iterations to learn a good opponent strategy. It also does not reuse information learned from previous iterations (for the recommendation of a good opponent strategy). Our learned best response functions, as we note, are able to retain information from previous iterations, and moreover do not require an expensive optimization procedure on each iteration.
>
> As they note, their method is similar to that of Metz et al. (which we cite in our paper), though with the opponent initialized from scratch on each interaction and with finite-time convergence guarantees.
>
> Despite those major differences, the Fiez et al. paper is relevant to ours and we have added a citation to it. We thank the reviewer for bringing it to our attention.
>
> "The experimental settings are clearly described although code (or at least code samples) would help with reproducibility."
>
> Our full codebase for this paper overlaps with other papers. We intend to disentangle and release it as soon as possible. In the meantime, we have added some code to the appendix. If the reviewer would like to see other specific pieces of code, please let us know.

---

> > ### Comment · Reviewer_u8oJ · 2022-11-15
> > **I have increased my score to a 6**
> >
> > I appreciate the effort the authors took to respond to my questions. I have thus increased my score to 6. Some components of the author's answer could benefit from further improvement.
> >
> > Regarding non-zero sum games: Fiez et al's analysis does not seem to heavily depend on the fact that the game is zero sum. Their work reduces the original game to a nested optimization problem where the nested problem is linear to the opponent's optimization variables. Convergence in this class of problems has been thoroughly studied. Even if the game is not zero-sum, the utility of the opponent changes but remains linear so convergence properties are not affected.
> >
> > Regarding retention of information: Fiez et al's opponents indeed do not have memory. This is because every time an $x$ is submitted, the opponent finds the best linear combination as a response to $x$ that is independent of what the opponent played previously. If the opponent only did a single gradient step to update its linear weights (just like the b part does in this work) it would also look like it retains information across iterations (again just like this work). I am not sure if retention of information is specific to the author's framework.
> >
> > Regarding restrictions on best responses: Feiz et al's framework supports optimizing linear combinations over any algorithm that is differentiable over $x$, not just gradient descent algorithms. The work of the authors has far less restrictions because it does not focus on convergence guarantees. But differentiability of the best responses with $x$ is still the key property of the author's framework.
> >
> > It is clear that the framework of Feiz et al is much more narrow than this work. This to some extent is required in order to provide some convergence guarantees. The work of the authors, by focusing more on experimental validation, removes all these restrictions. I think it would be very helpful for the audience to understand how they can use this generality to their advantage.

---

### Official Review · Reviewer_5nGD · 2022-10-27

**Confidence:** 4
**Correctness:** 2
**Technical Novelty And Significance:** 3
**Empirical Novelty And Significance:** 2
**Recommendation:** 3

**Clarity, Quality, Novelty And Reproducibility:**

Clarity: In order to increase my score, I would strongly request from the authors to explain (step-by-step) the formulation of "full-fledged" model. Why it is well-defined? Why this is not a Stackelberg case follower framework?

Quality: Mediocre.

Novelty: Incremental (See above)

Extra comment: Optimistic Dynamics are dated by Popov et al and not Daskalakis et al.

**Details Of Ethics Concerns:**

Non applicable

**Strength And Weaknesses:**

In my humble opinion, this work is very incremental for the current venue. Since I want to give always the right of uncertainty, I will be really happy to read the answer of the authors in my objections.

First of all, the continuous dynamics may hide certainly the actual complexity of the problem since it could run in exponential time. Secondly, I don't understand how much easy is to compute the best-response at every round when I have multiple players ( I refer to the case where strategies belong to some convex sets, like simplex). It is not clear, if best-response for the case of multiple players correspond to the product of constrained sets (Nash equilibrium style) or in general optimization framework (Coarse Correlated Best Response style). Thirdly, how we are sure that the BR functional is continuous, smooth differentiable ?
How much easy is to compute such a Best-response for a given strategy? If we use an Poly TM to compute it, why this is differentiable?

Additionally, I don't understand the intuition about the descent in the functional. Can you explain me where the concept of 'exploitability' is hidden in this setting?

**Summary Of The Paper:**

This work proposes a new dynamical system (continuous dynamics) where the players follow  a gradient ascent style dynamics per player but in order to adapt fast to their opponents, each player assumes that all the other players follow their best-response in its own strategy. At the same time, best-response would evolve as a functional with a gradient descent dynamical system.
The strategy profile and best-response functions are trained simultaneously, with the former trying to minimize exploitability while the latter try to maximize it. They evaluate our method on various continuous games, showing that it outperforms prior methods.

**Summary Of The Review:**

In this work, the authors propose a new method for equilibrium finding based on the idea of learned best-response functions.

---

> ### Author Response · Authors · 2022-11-15
> **Response**
>
> We thank the reviewer for their feedback.
>
> "the continuous dynamics may hide certainly the actual complexity of the problem since it could run in exponential time"
>
> We are not exactly sure what is meant by this. Could the reviewer please clarify? In practice and in the experiments, we discretize the dynamics in time (see the second-last paragraph under the section "Algorithms"), just as is done for ordinary gradient descent.
>
> "I don't understand how much easy is to compute the best-response at every round when I have multiple players ( I refer to the case where strategies belong to some convex sets, like simplex)"
>
> The best response function takes the current strategy profile as input and yields best responses for each player. How "easy" it is to do this will depend on the form of the best response function. In the case of an affine best response function, like we use in this paper, it takes a single matrix-vector multiplication followed by an addition, where the number of dimensions is the number of parameters representing the players' joint strategy profile. For a low-rank affine best response function, the cost is even cheaper. However, as stated in the paper, the best response function could be more complex, such as a neural network.
>
> "It is not clear, if best-response for the case of multiple players correspond to the product of constrained sets (Nash equilibrium style) or in general optimization framework (Coarse Correlated Best Response style)."
>
> As stated in the paper, we are interested in finding a Nash equilibrium, that is, a strategy profile where exploitability (as defined on Page 2) is zero. We thus seek to minimize the exploitability function. The strategy profile space is a product of strategy spaces of the individual players. So, there is no correlation among players; the best-response function returns individual (uncorrelated) strategies for each player.
>
> "how we are sure that the BR functional is continuous, smooth differentiable ?"
>
> It need not be. As explained with our first example in the section "Single-player perspective", best responses can change discontinuously with respect to the current strategy profile. The discontinuous nature of best responses is one of the primary motivations for our method. Even though our particular choice of approximate best response function is continuous, while the true best response function may not be, the fact that it is a fully-fledged function allows it to adapt more quickly to changes in the strategy profile (and subsequent changes in true best responses). Our experiments support this.
>
> "I don't understand the intuition about the descent in the functional. Can you explain me where the concept of 'exploitability' is hidden in this setting?"
>
> On Page 2, we explain that the exploitability function is non-negative and zero precisely at Nash equilibria. Thus we seek to minimize (an approximation of) the exploitability function. We do this by performing gradient descent on (an approximation of) the exploitability function, much like how one seeks low-loss models in a supervised learning setting by performing gradient descent on the loss function, even when the latter might be nonconvex and have local minima.
>
> "I would strongly request from the authors to explain (step-by-step) the formulation of "full-fledged" model. Why it is well-defined? Why this is not a Stackelberg case follower framework?"
>
> We are not exactly sure what is meant by this. No player is privileged as a "leader" in our framework. Rather, we seek to find a Nash equilibrium by minimizing the (approximate) exploitability (as defined on Page 2) of the strategy profile. Since we do not always have access to (exact) best responses, we simultaneously try to learn a best response function that takes the current strategy profile as input and returns (approximate) best responses for each player.
>
> "Extra comment: Optimistic Dynamics are dated by Popov et al and not Daskalakis et al."
>
> We have added a citation to Popov et al.

---

### Decision · Program_Chairs · 2023-01-20

**Decision:**

Reject

**Justification For Why Not Higher Score:**

The authors propose an interesting idea, but it would need to be developed further before being accepted at a top-tier venue.

**Justification For Why Not Lower Score:**

N/A

**Metareview: Summary, Strengths And Weaknesses:**

This paper studies the problem of computing approximate Nash equilibria of games with continuous action spaces. To that end, the authors propose a scheme based on "learned best response" functions, intended to be a parametric proxy to the game's true best response functions. However, even though the authors claim to use a trained neural network to represent these functions, their experiments are actually run with affine proxies which have very limited expressive power in general continuous games. The experiments presented by the authors are individually linear (or close enough) for the most part, so this shortcoming is not present in the authors' validation campaign; in general however, this could be a significant roadblock, and it is not clear how one would actually train a neural network to learn such a best response function efficiently.

Because of the above, the conclusion of the discussion phase was that the paper does not meet the acceptance criteria for ICLR, so a decision was reached to make a "reject" recommendation to the program committee. The situation would be different if the paper provided a richer validation analysis or theoretical guarantees for a relevant class of games, but the current version of the paper is not yet there.

**Summary Of Ac-Reviewer Meeting:**

N/A